# Picky Eating in Children: A Scoping Review to Examine Its Intrinsic and Extrinsic Features and How They Relate to Identification

**DOI:** 10.3390/ijerph18179067

**Published:** 2021-08-27

**Authors:** Laine Chilman, Ann Kennedy-Behr, Thuy Frakking, Libby Swanepoel, Michele Verdonck

**Affiliations:** 1School of Health and Behavioural Sciences, University of the Sunshine Coast, Locked Bag 4 Maroochydore, Maroochydore DC, Sunshine Coast, QLD 4558, Australia ; ann.kennedy-behr@unisa.edu.au (A.K.-B.); lswanepo@usc.edu.au (L.S.); michele.verdonck@usc.edu.au (M.V.); 2School of Allied Health & Human Performance, University of South Australia, Adelaide, SA 5000, Australia; 3Research Development Unit, Caboolture Hospital, Metro North Hospital & Health Service, Herston, QLD 4510, Australia; thuy.frakking@health.qld.gov.au; 4Centre for Clinical Research, School of Medicine, The University of Queensland, Herston, QLD 4029, Australia

**Keywords:** environmental influences, picky eating, fussy eating, assessment, intrinsic features, extrinsic features

## Abstract

The health benefits and importance of family mealtimes have been extensively documented. Picky eating can impact this complex activity and has numerous extrinsic (or external) and intrinsic (or internal) features. Occupational therapists work with children and their families by looking at both intrinsic and extrinsic influences and are therefore well-placed to work within this context. This scoping review comprises a comprehensive search of key health industry databases using pre-determined search terms. A robust screening process took place using the authors pre-agreed inclusion and exclusion criteria. There were 80 studies that met the inclusion criteria, which were then mapped using content analysis. The most common assessments used to identify picky eating relied on parental reports and recall. Often additional assessments were included in studies to identify both the intrinsic and extrinsic features and presentation. The most common reported intrinsic features of the child who is a picky eater included increased sensitivity particularly to taste and smell and the child’s personality. Extrinsic features which appear to increase the likelihood of picky eating are authoritarian parenting, rewards for eating, and pressuring the child to eat. Most commonly reported extrinsic features that decrease the likelihood of picky eating are family meals, responsive parents, and involving the child in the preparation of food. In conclusion, there is a lack of published papers addressing the role of occupational therapists in the assessment and identification of picky eating in children. There appears to be a complex interplay between intrinsic and extrinsic features which impact caregiver responses and therefore on the picky eater.

## 1. Introduction

The World Health Organization recommends that in addition to an adequate variety, amount, and frequency of foods, feeding times should be periods of learning and love [1]. Consuming a wide variety of foods, including items such as vegetables and fruit, has been established as important for health and has been shown to reduce the risk of chronic diseases [2]. One reason for the disruption of eating and feeding is picky eating. Ekstein [3] defines picky eating as a child’s unwillingness to eat familiar or new foods that is severe enough to interfere with daily routines and the parent-child relationship. This definition suggests that picky eating results from a combination of intrinsic (client-specific factors such as personality, body functions, and structures) and extrinsic (those that are external such as the environment and others within it) factors. These thereby impact participation in everyday activities that are related to mealtimes. Occupational therapists are health professionals who consider all aspects of the person, including psychosocial, cultural, and environmental factors when considering impairments [4]. The act of self-feeding is seen as a complex activity and numerous issues can disrupt its execution [5]. It is also considered a co-activity in the early years of a child’s life, that is one that requires active participation by both the caregiver and the child [6]. Feeding is defined within the Occupational Therapy Practice Framework as setting up, arranging, and bringing food or fluid from the vessel to the mouth and includes self-feeding and feeding others, whereas the occupation of eating is defined as keeping and manipulating food or fluid in the mouth [6]. Feeding and eating are central to the framework of family life and are strongly embedded within culture and tradition [7,8]. In Australia, family mealtimes are considered frequent and important [9]. As a child develops, they play a more active role in this process and are more able to exert control over what they eat. By the age of 24 months, a typically developing child is able to independently feed themselves and is able to use a fork and an open cup for drinking.

Discrepancies exist between the parental level of concern regarding picky eating and the health professional response. A contradiction may exist as caregivers have reported worrying about their child’s limited intake even when they were reassured by health professionals that their child was in good health [10]. Frustrations were identified when health professionals suggested that a child who is a picky eater needed to increase their intake without suggestions about how to do this [11,12]. Early identification and intervention are important to support parents, as research has indicated that mothers, with children who have feeding problems, have higher levels of depression and anxiety [13]. Occupational therapists are in the unique position to work with the client and their family to decipher what occupational roles are impacted by picky eating and therefore how the occupations are impacted. They can provide intervention for feeding-related issues in children with positive outcomes in the areas of feeding performance, feeding interaction, and feeding competence of parents and children [4].

Previous systematic reviews regarding picky eating have been limited by variable definitions of paediatric picky eaters with subsequently varied prevalence [14,15,16]. These systematic reviews did not capture the complexity of different aspects of picky eating or consider the intrinsic and extrinsic features together. The purpose of this study was to perform a scoping review on the intrinsic and extrinsic features of picky eating, the impact it has on the family, and to map the tools used to identify picky eating in children.

## 2. Materials and Methods

A scoping review was chosen due to emerging evidence on both intrinsic and extrinsic features of picky eating, whilst still ensuring a rigorous and transparent method for mapping [17]. Arksey and O’Malley’s [17] five-stage framework was applied, incorporating the suggestions of Khalil, et al. [18] and the checklist from the PRISMA extension for scoping reviews [19].

### 2.1. Stage 1: Identify the Research Questions

Given the complexity of the topic, four research questions were identified for this review:What are the intrinsic features of paediatric picky eaters? (e.g., the child’s personality);What are the extrinsic features of paediatric picky eaters? (e.g., others in the child’s environment such as parents);What tools or methods have been used to identify paediatric picky eaters?What is the impact of picky eating on others around them (i.e., extrinsic impacts)?

### 2.2. Stage 2: Identifying Relevant Studies

A search was conducted of Scopus, Web of Science, and CINAHL between June 2020 and August 2020 using the following keywords:

Picky (picky, fuss*, faddy, choosy, selective) AND (2) Eating (eat*, feed*, drink*, self-feed*) AND (3) child (Child*, paediat*, pediat* todd*).

### 2.3. Stage 3: Study Selection

Inclusion criteria were as follows: Published between January 2010–January 2020

Children aged 2–10 years (or families/caregivers of these children);Studies with a focus and definition of paediatric picky eating or other similar labels including fussy eating/choosy eating/faddy eating/selective eating or food fussiness;Interventions of picky eating focused on what impacts behaviours.

Two reviewers independently examined the title and abstract against the inclusion criteria to determine the eligibility of inclusion of the study with any uncertainty initially resulting in inclusion. Please see the modified PRISMA flow diagram (Figure 1). Any discrepancies were discussed and reviewed using the exclusion and inclusion criteria to resolve these.

The search yielded a total of 772 papers. After the removal of duplicates, 438 records remained. Following the screening of titles and abstracts by two independent reviewers, this was reduced to a total of 83 papers.

### 2.4. Stage 4: Charting the Data

Full texts were retrieved and read to ascertain relevance to the research questions. Two reviewers analysed each paper independently and recorded details including the author, year of publication, location of the study, study type/methodology, study aims/purpose, study population and study size, the definition of picky eating, child characteristics and behaviours associated with picky eating, physical and social environmental influences, and assessment tool used to identify picky eating. 

### 2.5. Stage 5: Collating, Summarising and Reporting Results

The evidence was mapped using content analysis according to the above headings, and will be reported in the following section.

## 3. Results

### 3.1. Study Characteristics

Cross sectional studies using parental reports of the child in a single point of time, often using multiple assessments were the most common study type [20,21,22,23,24,25,26,27,28,29,30,31,32,33,34,35,36,37,38,39,40,41,42,43,44,45,46,47,48,49,50,51,52,53,54,55,56,57,58,59,60]. Longitudinal studies gathering data, again usually parental report, at different time points were also common [61,62,63,64,65,66,67,68,69,70,71,72,73,74,75]. Ages and demographics of target population of included children also greatly varied between studies, with no consistency between age brackets considered or size of the age bracket. Most commonly the age bracket of toddlers (12 to 36 months old) or preschool aged children (aged three to five) was included either as the primary focus [20,25,30,34,36,37,38,39,41,44,52,57,59,60,72,73,76,77,78,79,80,81] or as part of a larger age inclusion [21,23,24,28,35,40,42,43,47,49,51,63,64,68,69,71,74,75,82,83,84,85,86,87] or a focus on primary school aged (five to eleven years) [27,31,32,48,53,54,55,56,61,70,88]. Another common focus was that of children from a socioeconomically disadvantaged area [20,33,53,58,62,82].

### 3.2. Assessments Used to Identify Picky Eating 

The most frequently use tool to identify picky eating was the food fussiness subscale of the Children’s Eating Behaviour Questionnaire (CEBQ) [21,22,27,30,32,35,37,42,44,54,55,60,65,69,72] and other variants of the CEBQ [23,29,33,45,53,56,61,80,82,89,90]. See Table 1.

### 3.3. Assessment Tools Used to Measure Environmental Features 

Maternal depressive symptoms were measured by the Patient Health Questionnaire (PHQ) [62] the Brief Symptom Inventory (BSI) [69], the Hopkins Symptom Checklist (SCL-25) [81], or the Centre for Epidemiologic Studies-Depression scale [35]. Maternal perceptions of parent-child interaction were measured using the Parenting Stress Index (short form) [62].

Other studies used the Caregiver’s Feeding Styles Questionnaire (CFSQ) which classified parents into different feeding styles such as authoritarian or uninvolved [33,37,53], or most commonly the Feeding Practices and Structure Questionaire-2 (FPSQ) that examined mealtimes structure [34,42,60,65]. Mealtime characteristics were also investigated as separate constructs and used questions such as the Family Ritual Questionnaire [64] or the Mealtime Assessment Survey [52]. See Table 2.

### 3.4. Intrinsic Assessments-Those That Are Specific to the Child 

Results show that a multitude of assessment tools exist to identify aspects of picky eating, which are often used in combination to provide a holistic profile of the child and their environment. See Table 3.

### 3.5. The Child Who Is a Picky Eater: Presentation

Several features of the child’s presentation were linked to a possible consideration identification of picky eating. The most cited feature was reduced intake by the child, in particular reduced intake, or reduced preference for vegetables [14,22,44,47,48,58,78,79,80,86,91] reduced variety in general [15,47,86,92] or refusing food based on texture [47,78,86]. Other consistent features in the presentation of children with picky eating tried fewer foods or rejecting new foods [15,40,52,75,79,86,91,92] and having a smaller range of liked food [15,75].

Behaviours of the child who was identified as a picky eater included slower to eat or longer feeding time than children who are typically developing [15,29,52,79,86], avoiding mealtimes [52,79,92] or sitting at the table [20,52], and rigid behaviours regarding food [52,79] such as inspecting foods [14,52]. Common responses of the picky eater to food included verbally [52] or physically gesturing refusal [14,37,75,79] or general disinterest in food [92].

### 3.6. The Child Who Is a Picky Eater: Intrinsic Features 

Increased sensitivity was the most cited feature of a picky eaters, particularly in relation to taste and smell [22,28,47,51,54,66,79,80,92] or to texture [27]. The picky eater was more often male [26,32,35,45,63,70,80], firstborn [63,74,78] and underweight [48,70,80,81,84]. However, the link to weight was unclear with some studies reporting no link between weight and picky eating factors [16,21,60,71], as was the link between gender and picky eating [22,75]. See Figure 2.

A child’s personality appears to impact the likelihood of picky eating. In particular, certain features of a child’s temperament were linked with an increased likelihood of picky eating, such as increased levels of frustration [35,36,61,81] or rigidity [27]. Similarly, an increased history of medical illness, feeding problems in infancy, or preterm delivery increased picky eating incidence [36,43,71,73] as did a secondary diagnosis such as Autism Spectrum Disorder (ASD) or Attention deficit hyperactivity disorder (ADHD) [22,56,80].

### 3.7. Paediatric Picky Eating–Extrinsic Features

Most common identified extrinsic influencer that increased the child’s likelihood of being a picky eater was maternal depressive or anxiety symptoms [25,62,63,69,71,74,79,81]. Other extrinsic features included mothers who had low food intake or were picky eaters themselves [63]. Some studies had conflicting data reported such as reduced maternal education levels [26,32,48,58,78,80] opposed to increased maternal education levels [15,24,45,63,86]. Other features of caregivers included younger parental age [43,48,80], low parental income [74,80], low parental weight [48], parental drug used [93] or parental sensory sensitivity [66].

### 3.8. Parenting Styles/Behaviours

Commonly reported was the parenting style used that increased likelihood of picky eating in particular authoritarian parenting [24,33,37,38,40,55] or unstructured [66] or uninvolved parenting [33]. Parenting styles related to feeding practices in particular that increased the likelihood of picky eating included offering a reward for eating [15,42,46,55,60,65,85] or persuasive feeding [65,85]. Significantly the parental response most often tied to increased picky eating was pressuring the child to eat [14,15,20,23,28,46,55,70,76,79,82,86,91]. However, this was also reported as bi-directional, for example, if caregivers perceived their child as picky, they were more likely to pressure their child to eat [53,59,60]. Further, it was reported that caregivers reported they are more likely to pressuring a child to eat if the child is underweight [14].

Physical environmental impacts that were linked to picky eating in children included television during meals [60,77] or early introduction of solids [15,57,63].

### 3.9. Extrinsic Influences That Decrease Likelihood of Picky Eating 

Conversely, there are environmental features that decreased the likelihood of picky eating in children. Caregivers eating with their children was the most cited protective feature [15,29,32,42,63,64,65,77]. This is similarly supported by caregivers setting a good example such as eating a variety of [70,91] healthy foods [25,63,64,90]. Parenting style also has an impact on reducing the likelihood of picky eating such as responsive parenting [15,33,42], using structured responses to food such as set mealtimes [42,65] autonomy promoting parenting [64], or avoiding non-responsive responses to eating such as pressure to eat [60]. See Figure 3.

Involving the child in the preparation of food was found to reduce the likelihood of picky eating [14,25,38,64], as was the child having choice over food [77] and having repeated exposure to healthy foods [28,72]. The presence of siblings was also reported to reduce picky eating [81] and increase the enjoyment of food [45]. Breastfeeding for greater than six months has also been shown to reduced picky eating [32,57]. An increase in physical activity levels reduces picky eating [26,36]. These results indicate that the environment surrounding the child can have a protective mechanism in reducing the likelihood of picky eating.

### 3.10. Impact of Picky Eating on Others

Picky eating behaviours have an impact on others around the child, in particular, the caregivers. The most commonly reported impact was caregivers choosing to prepare an alternative meal with a specific presentation for the child who is a picky eater [14,20,52,75,79,83,92].

Negativity towards mealtime for the caregiver, including emotions for the caregiver such as frustration [14,15,56], concern about wasting food [14,83], or the child’s weight [15]. Conflict between the caregiver and the child is often reported [75,92,93] with increased reports of stress for the caregiver [15,83,94] further impacting the problematic parent child interaction in a bi-directional manner [79] with such problematic relationships impacting the general climate at mealtimes [31].

Reward to eat was mentioned above as an extrinsic factor that increased picky eating, however, it was also reported as a response to picky eating [14,85] indicating a bi-directional influence regarding cause and effect. These results show that picky eating has negative impacts on others within their environment.

## 4. Discussion

The aim of the scoping review was to examine identified extrinsic features and intrinsic features of picky eating, and how they present and impact others around them. A further aim was to summarise current assessment tools used to identify a child as a picky eater.

The most commonly used tool to determine the presence of picky eating is the CEBQ, in particular the food fussiness subscale [95]. This assessment tool was designed to gather information about eight different feeding styles of children and not specifically for picky eating. Other methods of identification of picky eating included many studies asking singular questions of caregivers such as if they considered their child to be a picky eater [29,33,45,53,61,80,82,90]. This itself requires the caregiver to understand what picky eating is or the developmental appropriateness of the picky eating itself. Very few assessment tools incorporated expert opinion into the diagnostic process but relied on caregiver interpretation of the behaviours. Picky eating in children is considered by many as developmentally normal, so it’s important to determine between problematic and non-problematic picky eating [96].

Our most important finding was that there are identifiable intrinsic characteristics of the child that are linked to picky eating, that is, with characteristics about a child, such as personality, which cannot be readily changed if at all. To date, these include increased sensory sensitivity [22,27,28,47,51,54,66,79,80,92], certain personality or temperament traits such as rigidity [35,36,61,81], or developmental diagnosis [22,56,80]. Links to weight and gender were inconsistent with some studies reporting conflicting data. Features that are potentially modifiable, included reduced variety intake of certain foods such as vegetables, slower to eat, and avoidance of mealtimes.

A second important finding of the scoping review identified extrinsic or environmental features which increased the likelihood of picky eating. Maternal mental health, including depression or anxiety, eating habits of the mother including picky eating or reduced intake all increase the likelihood of picky eating. A strong relationship between increased picky eating in children and parenting style was reported in the literature, most commonly authoritarian parenting [24,33,37,38,40,55] or offering a reward for eating [15,42,46,55,60,65,85]. Conversely, parenting style was attributed to reducing the likelihood of picky eating including responsive parenting [15,33,42], structuring mealtimes [42,65], and autonomy promoting parenting [64]. Other extrinsic features appeared to reduce the likelihood of picky eating, such as family mealtimes [52,79,92] and involving the child in meal preparation [14,25,38,64]. What is interesting from this scoping review was that often other tools were required to measure other features about the child, such as sensory profiles [47,51] or their environment included people within this environment, some features of which are linked to influencing picky eating. This concern is supported by Pocock [97] who reported that health promotion strategies were best integrated when targeting the whole family and therefore the need for family-centered intervention.

The cause-and-effect description of picky eating is unclear. This is demonstrated in parenting response patterns, namely pressure to eat. Pressuring a child to eat was highly correlated with increased rates of childhood picky eating [53,59,60]. However, pressuring a child to eat was also reported as a response pattern to a child who was perceived as a picky eater or perceived as underweight [14]. It was therefore unclear regarding the cause- did picky eating cause the pressured feeding or vice versa. These findings are consistent with those of Cardona Cano, Hoek, and Bryant-Waugh [93] who suggested parental pressure to eat could be a reaction to a child’s picky eating or could result in increasing picky eating behaviour by reducing the enjoyment of food. It also corroborates the ideas of Wolstenholme, Kelly, Hennessy, and Heary [14] who suggested that parents and children change their behaviours in response to each other.

This unclear cause and effect are again seen in impacts on caregivers. There appears to be a problematic circular effect of picky eating- with picky eating been linked to parental stress, higher levels of parental stress were also linked to an increased likelihood of picky eating [15,83,94]. Stress itself then impacts the caregiver-child relationship at mealtimes which then impacts the atmosphere at mealtimes. In addition, mealtimes are linked to reduced picky eating, however, what can be done when the mealtime itself is causing the stress and picky eaters themselves are shown in the literature to avoid mealtimes. This finding is in agreement with Cardona Cano, Hoek, and Bryant-Waugh [93] who suggested that problematic parent-child interactions can exist with caregivers of picky eaters, suggesting a reconceptualization of picky eating to understand this bi-directional nature of picky eating [98].

Caregivers further report feeling unsupported by healthcare professionals [10]. Picky eating is linked to increased stress for the caregiver with again a circular effect with alterations in a caregiver’s mental health, such as depression or anxiety, linked with increased rates of picky eating. Such complex interplays would be difficult to accurately capture within a single assessment tool, suggesting a different way to capture information is required.

## 5. Limitations

This review excluded reports in languages other than in English, which potentially limited information on different cultural contexts. Another limitation was that studies that investigated picky eating as secondary to other diagnoses were not included, however, this was done to ensure homogeneity of the population. This review of picky eating considered a large range of ages of children, with many studies investigating picky eating with a similar range. As a scoping review is reliant upon the data reported in other studies and does not have access to raw data, specific ages could not be considered. In addition, the current scoping review is based on a wide age range between 2 to 10 years. Whilst inclusion of children 2 to 10 years old is similar to previously published studies in picky eating, we acknowledge that the wide age range increases the heterogeneity of paediatric feeding skills included in our review.

Further, a scoping review includes studies that were mixed in approach and quality with some using qualitative methodology, others standardised assessments, and others using non-stardardised tools. This limits conclusive comparisons between studies. This scoping review identifies the wide variability of results and findings of picky eating but as such is beyond the scope of metasysthesis and/or a metanalysis.

## 6. Conclusions

The aim of this study was to review the current evidence of both intrinsic and extrinsic features of picky eating. To the author’s knowledge, this scoping review is the first to consider both these features simultaneously.

This scoping review of 80 papers has confirmed that picky eating involves a complex interplay between intrinsic and extrinsic features reported using a variety of methodologies and outcome measures. Intrinsic and extrinsic features impact the presentation of picky eating as well as the caregiver response. This review indicates the importance of the environment, in particular mealtimes and parenting style. It identified that certain responses to picky eating, such as pressure to eat, are linked to increasing the likelihood of picky eating. Identifying these responses at the point of assessment may help guide the modification of the environmental influences on picky eating in children. It may be possible to alter the environmental influences while in contrast some intrinsic factors cannot be easily altered, such as personality or a child’s diagnosis. This highlights the opportunity for caregivers to understand environmental influences. Practice implications, therefore, indicate the need for family-centered care.

Intrinsic features, although more difficult to change, are important at the point of diagnosis and as such should be incorporated into assessment as currently there is a lack of validated assessment tools specific to picky eating in children that capture this. Picky eating is linked with certain behaviours of the child such as reduced intake, refusing food based on texture, or rejecting new foods. This review demonstrated that many picky eating tools have been used as well as additional non-specific picky eating assessments needed to understand the complete picky eater profile. A more holistic standardised tool that captures both the intrinsic and extrinsic and impacts on others is required so that picky eating can be more readily and consistently identified. As such, holistic assessments which include parent-child interactions associated with picky eating in children are required to inform evidence-based interventions that are effective.

Of concern is the impact a child who is a picky eater has on others around them, resulting not only in practicalities such as increased meal preparation time but also increased stress and a potentially problematic relationship between caregiver and child. There is a need for further research to investigate these problematic parent-child interactions as a result of picky eating. Assessment of picky eating should not be considered in isolation but in conjunction with treatment as part of the information-gathering process, with intrinsic and extrinsic features still relevant for both assessment and treatment. Further investigation is needed regarding how these features are considered in practice.

For young children, self-feeding is considered a co-occupation. and mealtimes themselves are considered central to family life, therefore it is important to equally consider the impact not only on the child but also the caregiver. Despite the less concerned viewpoint of the health professional, which is often made based on weight or nourishment, there remains an increased report of stress of the primary caregiver of the picky eater. Further investigation into the features of picky eating is required based on both parent perception and expert opinion to improve the accuracy of identification and understanding of impact.

## Figures and Tables

**Figure 1 ijerph-18-09067-f001:**
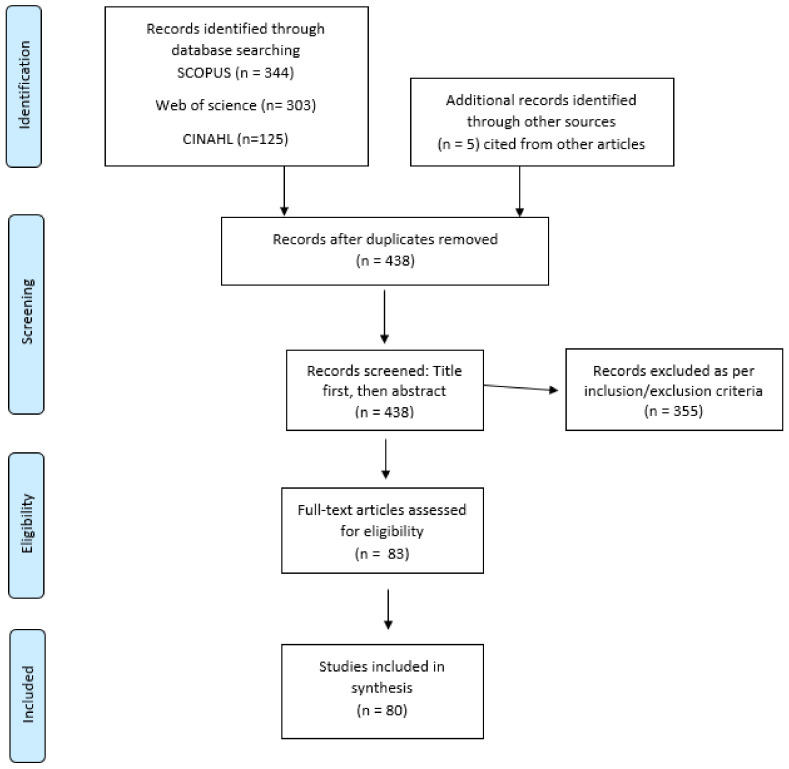
PRISMA Flow Diagram.

**Figure 2 ijerph-18-09067-f002:**
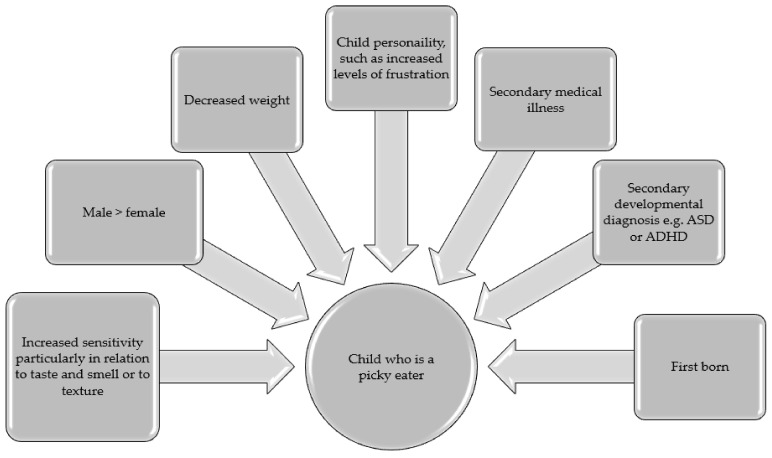
The child who is a picky eating: intrinsic features.

**Figure 3 ijerph-18-09067-f003:**
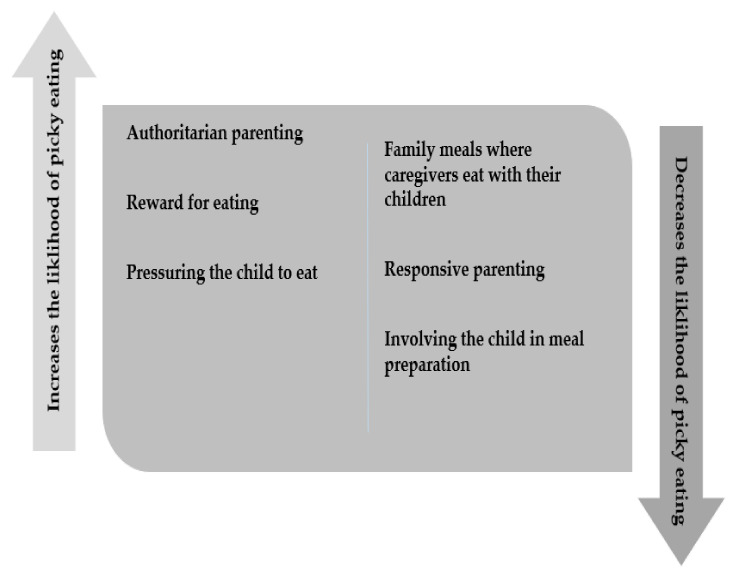
Extrinsic influences that increase or decrease picky eating.

**Table 1 ijerph-18-09067-t001:** Assessment tools used to measure or identify intrinsic features of picky eating.

Assessment Name	Brief Description	Study
Children’s Eating Behaviour Questionnaire (CEBQ): food fussiness subscale	Six questions thought to relate to food pickiness including: My child enjoys tasting new food; My child enjoys a wide variety of food; My child is interested in tasting food tasted before; My child refuses new foods at first; My child decides that she/he doesn’t like food, even without tasting it; My child is difficult to please with meal. Parents respond to each question on a 5-point scale (1 = never, 5 = always) and responses are averaged	[21,22,27,30,32,35,37,42,44,54,55,60,65,69,72]
Children’s Eating Behaviour Questionnaire (CEBQ)	A 35 itemed assessment thought to measure children eating behaviours. It has eight subscales: food responsiveness; good fussiness (as above); emotional overeating; enjoyment of food; desire to drink; satiety-responsiveness; slowness in eating and emotional undereating. Parents respond to each question on a 5-point scale (1 = never, 5 = always) and responses are averaged.	[29,33,45,53,61,80,82,90]
One single study specific question	For example, ‘is your child a picky eater’ or ‘is your child choosy about food’.	[29,33,45,53,61,80,82,90]
Study specific structured interview	Varied	[31,36,41]
Behaviour checklist: two items	These two items are thought to assess picky eating: these being lack of appetite and food fussiness. These items were scored 1–3 and then averaged.	[81]
Colorado Children’s Temperament Inventor: reaction to food subscale	This subscale asked the mother to rate her child on a series of 5 items about the child’s typical responses to food and includes items common across picky-eating scale.	[62]
Behavioural Paediatric Feeding Assessment Scale (BPFAS): frequency subscale	This subscale is used to measure food avoidance. Parents complete thirty-five items on a five-point Likert scale.	[29]
Preschool Age Psychiatric Assessment (PAPA)	A semi-structured psychiatric interview for parents including questions about child’s food preferences and appetite over previous three months, including restriction of feeds and whether food selectivity impairs functioning.	[66,79]
Child Behaviour Checklist: picky eating subscale	Uses two questions from the 99 itemed parent report questionnaire: ‘does not eat well’ and ‘refuses to eat’ on a three-point Likert scale	[43,74]
Mealtime assessment survey (MAS): singular picky eating question	Contains 43 items about toddler mealtime behaviours and 25 items focused on parent mealtime strategies; within the study children were categorised as picky or non-picky by one question: ‘is your child a picky eater’.	[38]
Brief Assessment of Mealtime Behaviour in Children (BAMBIC): subscale	10-item parent report questionnaire on mealtime behaviour. Raw subscale scores for Limited Variety, Food Refusal, and Disruptive Behaviour were included as variables in this study.	[47]
Food Preference Questionnaire for children (FPQ)	Caregivers rate their child’s liking for 75 commonly consumed individual foods from six food groups. Items are scored on a five-point Likert scale or a never tried option which is scored as a missing response.	[22]
Child-reported Food Preference Questionnaire (C-FPQ) (based on the FPQ)In pilot phase for Australian children aged 7–12 years old	Child’s food preferences to core foods: Fruits, vegetables, protein-rich (meat and alternatives, breads, etc.), carbohydrate-rich (cereals, etc.), dairy and non-core sweet and savory foods and beverages. Measured on a 5-point Likert scale.	[88]
Picky eating questionnaire (PEQ) In pilot phase for Australian children aged 7–12 years old	Measures primary caregiver’s perception of their child’s pickiness to core foods: fruits; vegetables; meat and alternatives; breads and cereals; dairy. Measured on a 5-point Likert scale.	[88]
Oregon Research Institute Child Eating Behaviour Inventory (ORI-CEBI)	An items pool addressing child eating behaviours outcome of picky eating behaviour was assessed using certain subscales limited variety, preparation for food preparation and food refusals.	[16,57,64]

**Table 2 ijerph-18-09067-t002:** Assessment tools used to measure environmental features.

Feature Measured	Assessment Name	Study
Maternal depressive symptoms	Patient Health Questionnaire (PHQ)	[62]
	Brief Symptom Inventory (BSI)	[69]
	Hopkins Symptom Checklist (SCL-25)	[81]
Parents different feeding styles and feeding practices	Caregiver’s Feeding Styles Questionnaire (CFSQ)	[33,37,53]
	Child Feeding Questionnaire	[70]
	Comprehensive Feeding Practices Questionnaire (CFPQ)	[64]
	Stanford feeding questionnaire	[67]
	Feeding Practices and Structure Questionaire-2 (FPSQ)	[34,42,60,65]
Family mealtime characteristics	Family Ritual Questionnaire	[64]
	Mealtime Assessment Survey	[52]
Maternal perceptions of parent-child interaction	Parenting Stress Index (short form)	[62]
Eating environment including parent feeding practices	Comprehensive feeding practices questionnaire	[23]
	Feeding practices and structures questionnaire (FPSQ)	[20]
Parental neophobia	Food Neophobia Scale (FNS)	[32]
Health and lifestyle including diet	Study specific questionnaire	[32]
Parent modelling during a mealtime	Parental Modelling of Eating Behaviours–Observational coding scheme (PARM-O) (study specific)	[49]
General parenting style	Parenting Styles and Dimensions Questionnaire (PSDQ)	[38]

**Table 3 ijerph-18-09067-t003:** Assessment used to measure other intrinsic features of the child (but not used as a diagnosis for picky eating in the study).

Assessment Tool	Brief Description	Study
Stanford feeding questionnaire	Parental report that measures child feeding behaviours such as limited variety of food, accepts new foods readily, strong likes/dislikes.	[67]
Children’s Behaviour Questionnaire short form	In this study was used to assess difficult temperament by using certain subscales	[35]
Child Mealtime Coding Scheme (CMCS)	Used to generate an overall index of how easy or difficult the child was to feed, ranging from 1 (easy; e.g., usually autonomous feeder, eats well with little protest) to 5 (difficult; e.g., much resistance to offers of food, refusal to eat).	[77]
Child Behaviour Checklist: full form	Provides information on eight empirically based syndromes: emotional reactivity, anxious/depressed, somatic complaints, withdrawn, sleep problems, attention problems, and aggressive behaviour.	[43,47,79,81]
Emotionality, Activity, Shyness and Sociability Temperament Survey (EAS)	Measures childhood temperament using 20 itemed survey.	[81]
Repetitive Behaviour Scale–Revised	A 43-item caregiver report measure that assesses restricted and repetitive behaviours in children.	[47]
Denver Developmental Screening Test 2	A tool to assess screen children’s development in four areas of functioning: fine motor-adaptive, gross motor, personal-social, and language skills.	[36]
Children’s Dietary Questionnaire (CDQ)	A parental report that assesses the child’s intake patterns.	[72]
Food Frequency Questionnaire (FFQ)	Assesses children’s food intake based on the frequency and type of food consumed over the past 4 weeks.	[47,74,80]
Short Sensory Profile (SSP)	Measures a child’s sensory responses including sensitivity to taste and processing tactile stimuli.	[47,51]

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
