# Peer review of "Picky Eating in Children: A Scoping Review to Examine Its Intrinsic and Extrinsic Features and How They Relate to Identification"

_ijerph, 2021, doi:10.3390/ijerph18179067_

Round 1

Reviewer 1 Report

To understand the picky eating in children due to extrinsic (or external) and intrinsic (or internal) features, this scoping review comprises a comprehensive search of key health industry databases using pre-determined search terms. I like to give the following comments.

  1. The picky eating cited from reference(s) in journal seems better than from a book (reference 3rd) as shown in the introduction section.
  2. Children with feeding problems may have higher levels of depression and anxiety. Additionally, the previous systematic reviews regarding picky eating have been limited. Those were the main rationale of this report. Conclusion indicated the importance of the environment, particularly the mealtimes and parenting style. Is it reached the main aim?
  3. Merits of five-stage framework in this scoping were not introduced in detail. Additionally, limitation of it was not conducted.
  4. The child’s personality as one of the intrinsic features seems ignored. Why?
  5. The picky eater was easier to identify using the extrinsic features. Is it included in the concern for conclusion?
  6. A problematic parent-child interaction can exist with caregivers of picky eaters, as mentioned in discussion section. How to improve it?

Reviewer 2 Report

Thanks for inviting me to review this paper, which is basically well written. The only concern that I may have is the inclusion age that ranges from 2 -10 years. Sometimes the behavioral presentations of picky eating in age 10 may significantly differ from that in age 2. I would suggest the authors tear apart the findings by age groups, such as age 2-5 years vs. 6-10 years. As such, the discussions may be more focused, and clinical implications may be more specified.  

Reviewer 3 Report

This article shows a scoping review about a very interesting topic, intrinsic and extrinsic features of picky eating in children. I think that this is a good research that is also difficult to perform, as many of the articles reviewed use qualitative evaluations. Thus, information management and obtaining conclusions are no easy tasks. The research is also ambitious, and therefore plenty of results are shown. My main concerns are the following:

— The Results Section is a bit dense and perhaps even a bit confusing. I think that this is generated by two main factors: the authors show many results; and, instead of referring the number of studies associated with the features shown, they show the actual references of those studies. Additionally, there is some repetition of the content of the tables and the main text. Therefore, I think that this section perhaps could be summarized a bit to facilitate an easier reading.

— The Discussion Section is easier to read and understand. Nevertheless, there are some parts that do not include references (for example, lines 267-293). For example, I think that after the sentences «A strong relationship between increased picky eating in children and parenting style was reported in the literature», or «The cause and effect description of picky eating is unclear», some articles should be cited.

— I humbly think that this research has more limitations than those showed in the article. For example, some of the aspects evaluated by the articles reviewed are qualitative, even if they have performed quantitative evaluations, using scales that perhaps have been validated, or not. This aspect is important and I think that perhaps could be mentioned, as this could limit the results and the conclusions.

— The limitations of the research seem to be in Conclusions Section.

— I think that the Conclusions Section do not show a clear and brief summary of the findings and the main conclusions obtained by the authors. The authors have done a great work in this review, that perhaps is not clearly summarized in this section (while it is better reflected in others, like Discussion).

Despite these aspects, that can be easily solved by the authors, I think that this is a good article that reflects a good and interesting research. I think that readability could be improved, specially in the Results Sections; that the Discussion Section could be also slightly improved, adding some references and better showing the limitations; and the Conclusions perhaps could better reflect the work performed by the authors. But overall, I really liked this article and I think that the authors have performed a great work.

Round 2

Reviewer 2 Report

Thanks for addressing my concerns. I have no more questions.